# Local Expectation Gradients for Black Box Variational Inference

**Michalis K. Titsias**
Athens University of Economics and Business
mtitsias@aueb.gr

**Miguel Lázaro-Gredilla**
Vicarious
miguel@vicarious.com

## Abstract

We introduce *local expectation gradients* which is a general purpose stochastic variational inference algorithm for constructing stochastic gradients by sampling from the variational distribution. This algorithm divides the problem of estimating the stochastic gradients over multiple variational parameters into smaller sub-tasks so that each sub-task explores intelligently the most relevant part of the variational distribution. This is achieved by performing an exact expectation over the single random variable that most correlates with the variational parameter of interest resulting in a Rao-Blackwellized estimate that has low variance. Our method works efficiently for both continuous and discrete random variables. Furthermore, the proposed algorithm has interesting similarities with Gibbs sampling but at the same time, unlike Gibbs sampling, can be trivially parallelized.

## 1 Introduction

Stochastic variational inference has emerged as a promising and flexible framework for performing large scale approximate inference in complex probabilistic models. It significantly extends the traditional variational inference framework [7, 1] by incorporating stochastic approximation [16] into the optimization of the variational lower bound. Currently, there exist two major research directions in stochastic variational inference. The first one (*data stochasticity*) attempts to deal with massive datasets by constructing stochastic gradients by using mini-batches of training examples [5, 6]. The second direction (*expectation stochasticity*) aims at dealing with the intractable expectations under the variational distribution that are encountered in non-conjugate probabilistic models [12, 14, 10, 18, 8, 15, 20]. Unifying these two ideas, it is possible to use stochastic gradients to address both massive datasets and intractable expectations. This results in a *doubly stochastic* estimation approach, where the mini-batch source of stochasticity can be combined with the stochasticity associated with sampling from the variational distribution.

In this paper, we are interested to further investigate the expectation stochasticity that in practice is dealt with by drawing samples from the variational distribution. A challenging issue here is concerned with the variance reduction of the stochastic gradients. Specifically, while the method based on the log derivative trick is currently the most general one, it has been observed to severely suffer from high variance problems [12, 14, 10] and thus it is only applicable together with sophisticated variance reduction techniques based on control variates. However, the construction of efficient control variates can be strongly dependent on the form of the probabilistic model. Therefore, it would be highly desirable to introduce more black box procedures, where simple stochastic gradients can work well for any model, thus allowing the end-user not to worry about having to design model-dependent variance reduction techniques. Notice, that for continuous random variables and differentiable functions the reparametrization approach [8, 15, 20] offers a simple black box procedure [20, 9] which does not require further model-dependent variance reduction. However, reparametrization is neither applicable for discrete spaces nor for non-differentiable models and this greatly limits its scope of applicability.

In this paper, we introduce a simple black box algorithm for stochastic optimization in variational inference which provides stochastic gradients having low variance and without needing any extra variance reduction. This method is based on a new trick referred to as *local expectation or integration*. The key idea here is that stochastic gradient estimation over multiple variational parameters can be divided into smaller sub-tasks where each sub-task requires different amounts of information about different parts of the variational distribution. More precisely, each sub-task aims at exploiting the conditional independence structure of the variational distribution. Based on this intuitive idea we introduce the *local expectation gradients* algorithm that provides a stochastic gradient over a variational parameter $v_i$ by performing an exact expectation over the associated latent variable $x_i$ while using a single sample from the remaining latent variables. Essentially, this consists of a Rao-Blackwellized estimate that allows to dramatically reduce the variance of the stochastic gradient so that, for instance, for continuous spaces the new stochastic gradient is guaranteed to have lower variance than the stochastic gradient corresponding to the reparametrization method where the latter utilizes a single sample. Furthermore, the local expectation algorithm has interesting similarities with Gibbs sampling with the important difference, that unlike Gibbs sampling, it can be trivially parallelized.

## 2 Stochastic variational inference

Here, we discuss the main ideas behind current algorithms on stochastic variational inference and particularly methods that sample from the variational distribution in order to approximate intractable expectations using Monte Carlo. Given a joint probability distribution $p(\mathbf{y}, \mathbf{x})$ where $\mathbf{y}$ are observations and $\mathbf{x}$ are latent variables (possibly including model parameters that consist of random variables) and a variational distribution $q_{\mathbf{v}}(\mathbf{x})$, the objective is to maximize the lower bound

$$\mathcal{F}(\mathbf{v}) = \mathbf{E}_{q_{\mathbf{v}}(\mathbf{x})} \left[ \log p(\mathbf{y}, \mathbf{x}) - \log q_{\mathbf{v}}(\mathbf{x}) \right], \tag{1}$$

$$= \mathbf{E}_{q_{\mathbf{v}}(\mathbf{x})} \left[ \log p(\mathbf{y}, \mathbf{x}) \right] - \mathbf{E}_{q_{\mathbf{v}}(\mathbf{x})} \left[ \log q_{\mathbf{v}}(\mathbf{x}) \right], \tag{2}$$

with respect to the variational parameters $\mathbf{v}$. Ideally, in order to tune $\mathbf{v}$ we would like to have a closed-form expression for the lower bound so that we could subsequently maximize it by using standard optimization routines such as gradient-based algorithms. However, for many probabilistic models and forms of the variational distribution at least one of the two expectations in (2) is intractable. Therefore, in general we are faced with the following intractable expectation

$$\widetilde{\mathcal{F}}(\mathbf{v}) = \mathbf{E}_{q_{\mathbf{v}}(\mathbf{x})} \left[ f(\mathbf{x}) \right], \tag{3}$$

where $f(\mathbf{x})$ can be either $\log p(\mathbf{y}, \mathbf{x})$, $-\log q_{\mathbf{v}}(\mathbf{x})$ or $\log p(\mathbf{y}, \mathbf{x}) - \log q_{\mathbf{v}}(\mathbf{x})$, from which we would like to efficiently estimate the gradient over $\mathbf{v}$ in order to apply gradient-based optimization.

The most general method for estimating the gradient $\nabla_{\mathbf{v}} \widetilde{\mathcal{F}}(\mathbf{v})$ is based on the log derivative trick, also called likelihood ratio or REINFORCE, that has been invented in control theory and reinforcement learning [3, 21, 13] and used recently for variational inference [12, 14, 10]. Specifically, this makes use of the property $\nabla_{\mathbf{v}} q_{\mathbf{v}}(\mathbf{x}) = q_{\mathbf{v}}(\mathbf{x}) \nabla_{\mathbf{v}} \log q_{\mathbf{v}}(\mathbf{x})$, which allows to write the gradient as

$$\nabla_{\mathbf{v}} \widetilde{\mathcal{F}}(\mathbf{v}) = \mathbf{E}_{q_{\mathbf{v}}(\mathbf{x})} \left[ f(\mathbf{x}) \nabla_{\mathbf{v}} \log q_{\mathbf{v}}(\mathbf{x}) \right] \tag{4}$$

and then obtain an unbiased estimate according to

$$\frac{1}{S} \sum_{s=1}^{S} f(\mathbf{x}^{(s)}) \nabla_{\mathbf{v}} \log q_{\mathbf{v}}(\mathbf{x}^{(s)}), \tag{5}$$

where each $\mathbf{x}^{(s)}$ is an independent draw from $q_{\mathbf{v}}(\mathbf{x})$. While this estimate is unbiased, it has been observed to severely suffer from high variance so that in practice it is necessary to consider variance reduction techniques such as those based on control variates [12, 14, 10].

The second approach is suitable for continuous spaces where $f(\mathbf{x})$ is a differentiable function of $\mathbf{x}$ [8, 15, 20]. It is based on a simple transformation of (3) which allows to move the variational parameters $\mathbf{v}$ inside $f(\mathbf{x})$ so that eventually the expectation is taken over a base distribution that does not depend on the variational parameters any more. For example, if the variational distribution is the Gaussian $\mathcal{N}(\mathbf{x}|\boldsymbol{\mu}, LL^{\top})$ where $\mathbf{v} = (\boldsymbol{\mu}, L)$, the expectation in (3) can be re-written as $\widetilde{\mathcal{F}}(\boldsymbol{\mu}, L) =$

$\int \mathcal{N}(\mathbf{z}|\mathbf{0}, I)f(\boldsymbol{\mu} + L\mathbf{z})d\mathbf{z}$ and subsequently the gradient over $(\boldsymbol{\mu}, L)$ can be approximated by the following unbiased Monte Carlo estimate

$$\frac{1}{S}\sum_{s=1}^{S}\nabla_{(\boldsymbol{\mu},L)}f(\boldsymbol{\mu} + L\mathbf{z}^{(s)}), \qquad (6)$$

where each $\mathbf{z}^{(s)}$ is an independent sample from $\mathcal{N}(\mathbf{z}|\mathbf{0}, I)$. This estimate makes efficient use of the slope of $f(\mathbf{x})$ which allows to perform informative moves in the space of $(\boldsymbol{\mu}, L)$. Furthermore, it has been shown experimentally in several studies [8, 15, 20, 9] that the estimate in (6) has relatively low variance and can lead to efficient optimization even when a single sample is used at each iteration. Nevertheless, a limitation of the approach is that it is only applicable to models where $\mathbf{x}$ is continuous and $f(\mathbf{x})$ is differentiable. Even within this subset of models we are also additionally restricted to using certain classes of variational distributions for which reparametrization is possible.

Next we introduce an approach that is applicable to a broad class of models (both discrete and continuous), has favourable scaling properties and provides low-variance stochastic gradients.

## 3 Local expectation gradients

Suppose that the $n$-dimensional latent vector $\mathbf{x}$ in the probabilistic model takes values in some space $\mathcal{S}_1 \times \ldots \mathcal{S}_n$ where each set $\mathcal{S}_i$ can be continuous or discrete. We consider a variational distribution over $\mathbf{x}$ that is represented as a directed graphical model having the following joint density

$$q_{\mathbf{v}}(\mathbf{x}) = \prod_{i=1}^{n} q_{v_i}(x_i|\text{pa}_i), \qquad (7)$$

where $q_{v_i}(x_i|\text{pa}_i)$ is the conditional factor over $x_i$ given the set of the parents denoted by $\text{pa}_i$. We assume that each conditional factor has its own separate set of variational parameters $v_i$ and $\mathbf{v} = (v_i, \ldots, v_n)$. The objective is then to obtain a stochastic approximation for the gradient of the lower bound over each variational parameter $v_i$.

Our method is motivated by the observation that each $v_i$ is influenced mostly by its corresponding latent variable $x_i$ since $v_i$ determines the factor $q_{v_i}(x_i|\text{pa}_i)$. Therefore, to get information about the gradient of $v_i$ we should be exploring multiple possible values of $x_i$ and a rather smaller set of values from the remaining latent variables $\mathbf{x}_{\backslash i}$. Next we take this idea into the extreme where we will be using infinite draws from $x_i$ (i.e. essentially an exact expectation) together with just a single sample of $\mathbf{x}_{\backslash i}$. More precisely, we factorize the variational distribution as $q_{\mathbf{v}}(\mathbf{x}) = q(x_i|\text{mb}_i)q(\mathbf{x}_{\backslash i})$, where $\text{mb}_i$ denotes the Markov blanket of $x_i$. The gradient over $v_i$ can be written as

$$\nabla_{v_i}\widetilde{\mathcal{F}}(\mathbf{v}) = \mathbb{E}_{q(\mathbf{x})}\left[f(\mathbf{x})\nabla_{v_i}\log q_{v_i}(x_i|\text{pa}_i)\right] = \mathbb{E}_{q(\mathbf{x}_{\backslash i})}\left[\mathbb{E}_{q(x_i|\text{mb}_i)}\left[f(\mathbf{x})\nabla_{v_i}\log q_{v_i}(x_i|\text{pa}_i)\right]\right], \qquad (8)$$

where in the second expression we used the law of iterated expectations. Then, an unbiased stochastic gradient, say at the $t$-th iteration of an optimization algorithm, can be obtained by drawing a single sample $\mathbf{x}_{\backslash i}^{(t)}$ from $q(\mathbf{x}_{\backslash i})$ so that

$$\mathbb{E}_{q(x_i|\text{mb}_i^{(t)})}\left[f(\mathbf{x}_{\backslash i}^{(t)}, x_i)\nabla_{v_i}\log q_{v_i}(x_i|\text{pa}_i^{(t)})\right] = \sum_{x_i}\widetilde{q}(x_i|\text{mb}_i^{(t)})f(\mathbf{x}_{\backslash i}^{(t)}, x_i)\nabla_{v_i}q_{v_i}(x_i|\text{pa}_i^{(t)}), \qquad (9)$$

where $\sum_{x_i}$ denotes summation or integration and $\widetilde{q}(x_i|\text{mb}_i^{(t)})$ is the same as $q(x_i|\text{mb}_i^{(t)})$ but with $q_{v_i}(x_i|\text{pa}_i^{(t)})$ removed from the numerator.[1] The above is the expression for the proposed stochastic gradient for the parameter $v_i$. Notice that this estimate does not rely on the log derivative trick since we never draw samples from $q(x_i|\text{mb}_i^{(t)})$. Instead the trick here is to perform local expectation (integration or summation). To get an independent sample $\mathbf{x}_{\backslash i}^{(t)}$ from $q(\mathbf{x}_{\backslash i})$ we can simply simulate a full latent vector $\mathbf{x}^{(t)}$ from $q_{\mathbf{v}}(\mathbf{x})$ by applying the standard ancestral sampling procedure for directed graphical models [1]. Then, the sub-vector $\mathbf{x}_{\backslash i}^{(t)}$ is by construction an independent draw from the

**Algorithm 1** Stochastic variational inference using local expectation gradients

---
    **Input:** $f(\mathbf{x})$, $q_{\mathbf{v}}(\mathbf{x})$.
    Initialize $\mathbf{v}^{(0)}$, $t = 0$.
    **repeat**
       Set $t = t + 1$.
       Draw pivot sample $\mathbf{x}^{(t)} \sim q_{\mathbf{v}}(\mathbf{x})$.
       **for** $i = 1$ **to** $n$ **do**
          $dv_i = \mathbf{E}_{q(x_i|\mathrm{mb}_i^{(t)})}\left[f(\mathbf{x}_{\backslash i}^{(t)}, x_i)\nabla_{v_i} \log q_{v_i}(x_i|\mathrm{pa}_i^{(t)})\right]$.
          $v_i = v_i + \eta_t dv_i$.
       **end for**
    **until** convergence criterion is met.

---

marginal $q(\mathbf{x}_{\backslash i})$. Furthermore, the sample $\mathbf{x}^{(t)}$ can be thought of as a global or *pivot* sample that is needed to be drawn once and then it can be re-used multiple times in order to compute all stochastic gradients for all variational parameters $(v_1, \ldots, v_n)$ according to eq. (9).

When the variable $x_i$ takes discrete values, the expectation in eq. (9) reduces to a sum of terms associated with all possible values of $x_i$. On the other hand, when $x_i$ is a continuous variable the expectation in (9) corresponds to an univariate integral that in general may not be analytically tractable. In this case we shall use fast numerical integration methods.

We shall refer to the above algorithm for providing stochastic gradients over variational parameters as *local expectation gradients* and pseudo-code of a stochastic variational inference scheme that internally uses this algorithm is given in Algorithm 1. Notice that Algorithm 1 corresponds to the case where $f(\mathbf{x}) = \log p(\mathbf{y}, \mathbf{x}) - \log q_{\mathbf{v}}(\mathbf{x})$ while other cases can be expressed similarly.

In the next two sections we discuss some theoretical properties of local expectation gradients (Section 3.1) and draw interesting connections with Gibbs sampling (Section 3.2).

## 3.1 Properties of local expectation gradients

We first derive the variance of the stochastic estimates obtained by local expectation gradients. In our analysis, we will focus on the case of fitting a fully factorized variational distribution (and leave the more general case for future work) having the form

$$q_{\mathbf{v}}(\mathbf{x}) = \prod_{i=1}^{n} q_{v_i}(x_i). \tag{10}$$

For such case the local expectation gradient for each parameter $v_i$ from eq. (9) simplifies to

$$\mathbf{E}_{q_{v_i}(x_i)}\left[f(\mathbf{x}_{\backslash i}, x_i)\nabla_{v_i} \log q_{v_i}(x_i)\right] = \sum_{x_i} \nabla_{v_i} q_{v_i}(x_i)f(\mathbf{x}_{\backslash i}, x_i), \tag{11}$$

where also for notational simplicity we write $\mathbf{x}_{\backslash i}^{(t)}$ as $\mathbf{x}_{\backslash i}$. It would be useful to define the following mean and covariance functions

$$m(x_i) = \mathbf{E}_{q(\mathbf{x}_{\backslash i})}[f(\mathbf{x}_{\backslash i}, x_i)], \tag{12}$$

$$\mathrm{Cov}(x_i, x_i') = \mathbf{E}_{q(\mathbf{x}_{\backslash i})}[(f(\mathbf{x}_{\backslash i}, x_i) - m(x_i))(f(\mathbf{x}_{\backslash i}, x_i') - m(x_i'))], \tag{13}$$

that characterize the variability of $f(\mathbf{x}_{\backslash i}, x_i)$ as $\mathbf{x}_{\backslash i}$ varies according to $q(\mathbf{x}_{\backslash i})$. Notice that based on eq. (12) the exact gradient of the variational lower bound over $v_i$ can also be written as $\sum_{x_i} \nabla_{v_i} q_{v_i}(x_i)m(x_i)$, which has an analogous form to the local expectation gradient from (11) with the difference that $f(\mathbf{x}_{\backslash i}, x_i)$ is now replaced by its mean value $m(x_i)$.

We can now characterize the variance of the stochastic gradient and describe some additional properties. All proofs for the following statements are given in the Supplementary Material.

**Proposition 1.** The variance of the stochastic gradient in (11) can be written as

$$\sum_{x_i, x_i'} \nabla_{v_i} q_{v_i}(x_i)\nabla_{v_i} q_{v_i}(x_i')\mathrm{Cov}(x_i, x_i'). \tag{14}$$

This gives us some intuition about when we expect the variance of the estimate to be small. For instance, two simple cases are: i) when the covariance function $\text{Cov}(x_i, x_i')$ takes small values, which can occur when $q(\mathbf{x}_{\setminus i})$ has low entropy, or ii) when $\text{Cov}(x_i, x_i')$ is approximately constant. In fact, when $\text{Cov}(x_i, x_i')$ is exactly constant, then the variance is zero (so that the stochastic gradient is exact) as the following proposition states.

**Proposition 2.** If $\text{Cov}(x_i, x_i') = c$ for all $x_i$ and $x_i'$ then the variance in (14) is equal to zero.

A case for which the condition $\text{Cov}(x_i, x_i') = c$ holds exactly is when the function $f(\mathbf{x})$ factorizes as $f(\mathbf{x}_{\setminus i}, x_i) = f_i(x_i) + f_{\setminus i}(\mathbf{x}_{\setminus i})$ (see Supplementary Material for a proof). Such a factorization essentially implies that $x_i$ is independent from the remaining random variables, which results the local expectation gradient to be exact. In contrast, in order to get exactness by using the standard Monte Carlo stochastic gradient from eq. (5) (and any of its improvements that apply variance reduction) we will typically need to draw infinite number of samples.

To further analyze local expectation gradients we can contrast them with stochastic gradients obtained by the reparametrization trick [8, 15, 20]. Suppose that we can reparametrize the random variable $x_i \sim q_{v_i}(x_i)$ according to $x_i = g(v_i, z_i)$, where $z_i \sim q_i(z_i)$ and $q_i(z_i)$ is a suitable base distribution. We further assume that the function $f(\mathbf{x}_{\setminus i}, x_i)$ is differentiable with respect to $x_i$ and $g(v_i, z_i)$ is differentiable with respect to $v_i$. Then, the exact gradient with respect to the variational parameter $v_i$ can be reparametrized as

$$\nabla_{v_i} \left( \int q(\mathbf{x}_{\setminus i}) q_{v_i}(x_i) f(\mathbf{x}_{\setminus i}, x_i) d\mathbf{x} \right) = \int q(\mathbf{x}_{\setminus i}) q_i(z_i) \nabla_{v_i} f(\mathbf{x}_{\setminus i}, g(v_i, z_i)) d\mathbf{x}_{\setminus i} dz_i, \quad (15)$$

while a single-sample stochastic estimate that follows from this expression is

$$\nabla_{v_i} f(\mathbf{x}_{\setminus i}, g(v_i, z_i)), \quad \mathbf{x}_{\setminus i} \sim q(\mathbf{x}_{\setminus i}), \; z_i \sim q_i(z_i). \quad (16)$$

The following statement gives us a clear understanding about how this estimate compares with the corresponding local expectation gradient.

**Proposition 3.** Given that we can reparametrize $x_i$ as described above (and all differentiability conditions mentioned above hold), the gradient from (11) can be equivalently written as

$$\int q_i(z_i) \nabla_{v_i} f(\mathbf{x}_{\setminus i}, g(v_i, z_i)) dz_i, \quad \mathbf{x}_{\setminus i} \sim q(\mathbf{x}_{\setminus i}). \quad (17)$$

Clearly, the above expression is an expectation of the reparametrization gradient from eq. (16), and therefore based on the standard Rao-Blackwellization argument the variance of the local expectation gradient will always be lower or equal than the variance of a single-sample estimate based on the reparametrization method. Notice that the reparametrization method is only applicable to continuous random variables and differentiable functions $f(\mathbf{x})$. However, for such cases, reparametrization could be computationally more efficient than local expectation gradients since the latter approach will require to apply 1-D numerical integration to estimate the integral in (11) or the integral in (17)[2] which could be computationally more expensive.

### 3.2 Connection with Gibbs sampling

There are interesting similarities between local expectation gradients and Gibbs sampling. Firstly, notice that carrying out Gibbs sampling in the variational distribution in eq. (7) requires iteratively sampling from each conditional $q(x_i | \text{mb}_i)$, for $i = 1, \ldots, n$, and clearly the same conditional appears also in local expectation gradients with the obvious difference that instead of sampling from $q(x_i | \text{mb}_i)$ we now average under this distribution. Of course, in practice, we never perform Gibbs sampling on a variational distribution but instead on the true posterior distribution which is proportional to $e^{f(\mathbf{x})}$ (where we assumed that $-\log q_{\mathbf{v}}(\mathbf{x})$ is not part of $f(\mathbf{x})$). Specifically, at each Gibbs step we simulate a new value for some $x_i$ from the posterior conditional distribution that is proportional to $e^{f(\mathbf{x}_{\setminus i}^{(t)}, x_i)}$ and where $\mathbf{x}_{\setminus i}^{(t)}$ are the fixed values for the remaining random variables. We can observe that an update in local expectation gradients is quite similar, because now we also condition on some fixed remaining values $\mathbf{x}_{\setminus i}^{(t)}$ in order to update the parameter $v_i$ towards the direction

where $q(x_i|\text{mb}_i^{(t)})$ gets closer to the corresponding true posterior conditional distribution. Despite these similarities, there is a crucial computational difference between the two procedures. While in local expectation gradients it is perfectly valid to perform all updates of the variational parameters in parallel, given the pivot sample $\mathbf{x}^{(t)}$, in Gibbs sampling all updates need to be executed in a serial manner. This difference is essentially a consequence of the fundamental difference between variational inference and Gibbs sampling where the former relies on optimization while the latter on convergence of a Markov chain.

## 4 Experiments

In this section, we apply local expectation gradients (LeGrad) to two different types of stochastic variational inference problems and we compare it against the standard stochastic gradient based on the log derivative trick (LdGrad), that incorporates also variance reduction[3], as well as the reparametrization-based gradient (ReGrad) given by eq. (6). In Section 4.1, we consider a two-class classification problem using two digits from the MNIST database and we approximate a Bayesian logistic regression model using stochastic variational inference. Then, in Section 4.2 we consider sigmoid belief networks [11] and we fit them to the binarized version of the MNIST digits.

### 4.1 Bayesian logistic regression

In this section we compare the three approaches in a challenging binary classification problem using Bayesian logistic regression. Specifically, given a dataset $\mathcal{D} \equiv \{\mathbf{z}_j, y_j\}_{j=1}^m$, where $\mathbf{z}_j \in \mathbf{R}^n$ is the input and $y_j \in \{-1, +1\}$ the class label, we model the joint distribution over the observed labels and the parameters $\mathbf{w}$ by $p(\mathbf{y}, \mathbf{w}) = \left(\prod_{m=1}^M \sigma(y_m \mathbf{z}_m^\top \mathbf{w})\right) p(\mathbf{w})$, where $\sigma(a)$ is the sigmoid function and $p(\mathbf{w})$ denotes a zero-mean Gaussian prior on the weights $\mathbf{w}$. We wish to apply the three algorithms in order to approximate the posterior over the regression parameters by a factorized variational Gaussian distribution of the form $q_\mathbf{v}(\mathbf{w}) = \prod_{i=1}^n \mathcal{N}(w_i|\mu_i, \ell_i^2)$. In the following we consider a subset of the MNIST dataset that includes all 12660 training examples from the digit classes 2 and 7, each with 784 pixels so that by including the bias the number of weights is $n = 785$.

To obtain the local expectation gradient for each $(\mu_i, \ell_i)$ we need to apply 1-D numerical integration. We used the quadrature rule having $K = 5$ nodes[4] so that LeGrad was using $S = 785 \times 5$ function evaluations per gradient estimation. For LdGrad we also set the number of samples to $S = 785 \times 5$ so that LeGrad and LdGrad match exactly in the number of function evaluations and roughly in computational cost. When using the ReGrad approach based on (6) we construct the stochastic gradient using $K = 5$ target function gradient samples. This matches the computational cost, but ReGrad still has the unmatched advantage of having access to the gradient of the target function.

The variance of the stochastic gradient for parameter $\mu_1$ is shown in Figure 1(a)-(b). It is much smaller for LeGrad than for LdGrad, despite having almost similar computational cost and use the same amount of information about the target function. The evolution of the bound in Figure 1(c) clearly shows the advantage of using less noisy gradients. LdGrad will need a huge number of iterations to find the global optimum, despite having optimized the step size of its stochastic updates.

### 4.2 Sigmoid belief networks

In the second example we consider sigmoid belief networks (SBNs) [11] and i) compare our approach with LdGrad in terms of variance and optimization efficiency and then ii) we perform density estimation experiments by training sigmoid belief nets with fully connected hidden units using LeGrad. Note that ReGrad cannot be used on discrete models.

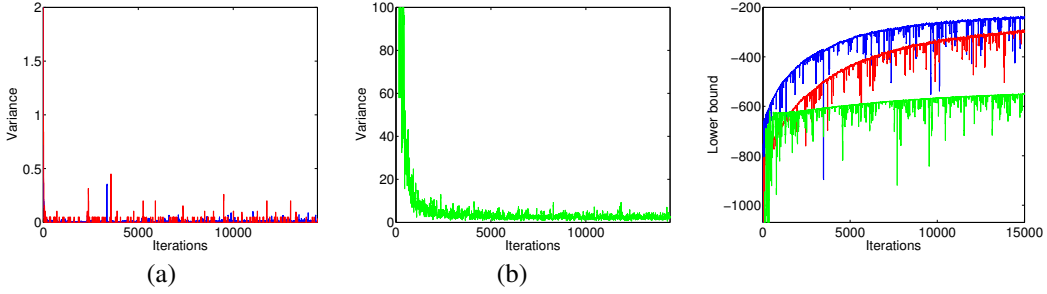

(a)　　　　　　　　　　(b)

Figure 1: (a) Variance of the gradient for the variational parameter $\mu_1$ for LeGrad (red line) and ReGrad (blue line). (b) Variance of the gradient for the variational parameter $\mu_1$ for LdGrad (green line). (c) Evolution of the stochastic value of the lower bound.

For the variance reduction comparison we consider a network with an unstructured hidden layer where binary observed vectors $\mathbf{y}_i \in \{0,1\}^D$ are generated independently according to

$$p(\mathbf{y}|W) = \sum_{\mathbf{x}} \prod_{d=1}^{D} \left[\sigma(\mathbf{w}_d^\top \mathbf{x})\right]^{y_d} \left[1 - \sigma(\mathbf{w}_d^\top \mathbf{x})\right]^{1-y_d} p(\mathbf{x}), \tag{18}$$

where $\mathbf{x} \in \{0,1\}^K$ is a vector of hidden variables that follows a uniform distribution. The matrix $W$ (which includes bias terms) contains the parameters to be estimated by fitting the model to the data. In theory we could use the EM algorithm to learn the parameters $W$, however, such an approach is not feasible because at the E step we need to compute the posterior distribution $p(\mathbf{x}_i|\mathbf{y}_i, W)$ over each hidden variable which clearly is intractable since each $\mathbf{x}_i$ takes $2^K$ values. Therefore, we need to apply approximate inference and next we consider stochastic variational inference using the local expectation gradients algorithm and compare this with the method in [19] eq. (8), which has the same scalability properties and have been denoting as LdGrad.

More precisely, we shall consider a variational distribution that consists of a recognition model [4, 2, 10, 8, 15] which is parametrized by a "reverse" sigmoid network that predicts the latent vector $\mathbf{x}_i$ from the associated observation $\mathbf{y}_i$: $q_V(\mathbf{x}_i) = \prod_{k=1}^{K} \left[\sigma(\mathbf{v}_k^\top \mathbf{y}_i)\right]^{x_{ik}} \left[1 - \sigma(\mathbf{v}_k^\top \mathbf{y}_i)\right]^{1-x_{ik}}$. The variational parameters are contained in matrix $V$ (also the bias terms). The application of stochastic variational inference boils down to constructing a separate lower bound for each pair $(\mathbf{y}_i, \mathbf{x}_i)$ so that the full bound is the sum of these individual terms (see Supplementary Material for explicit expressions). Then, the maximization of the bound proceeds by performing stochastic gradient updates for the model weights $W$ and the variational parameters $V$. The update for $W$ reduces to a logistic regression type of update, based upon drawing a single sample from the full variational distribution. On the other hand, obtaining effective and low variance stochastic gradients for the variational parameters $V$ is considered to be a very highly challenging task and current advanced methods are based on covariates that employ neural networks as auxiliary models [10]. In contrast, the local expectation gradient for each variational parameter $\mathbf{v}_k$ only requires evaluating

$$\nabla_{\mathbf{v}_k}\mathcal{F} = \sum_{i=1}^{n} \nabla_{\mathbf{v}_k}\mathcal{F}_i = \sum_{i=1}^{n} \sigma_{ik}(1 - \sigma_{ik}) \left[ \sum_{d=1}^{D} \log \frac{1 + e^{-\widetilde{y}_{id}\mathbf{w}_d^\top (\mathbf{x}_{i\backslash k}^{(t)}, x_{ik}=0)}}{1 + e^{-\widetilde{y}_{id}\mathbf{w}_d^\top (\mathbf{x}_{i\backslash k}^{(t)}, x_{ik}=1)}} + \log \frac{1 - \sigma_{ik}}{\sigma_{ik}} \right] \mathbf{y}_i, \tag{19}$$

where $\sigma_{ik} = \sigma(\mathbf{v}_k^\top \mathbf{y}_i)$ and $\widetilde{y}_{id}$ is the $\{-1,1\}$ encoding of $y_{id}$. This expression is a weighted sum across data terms where each term is a difference induced by the directions $x_{ik} = 1$ and $x_{ik} = 0$ for all hidden units $\{x_{ik}\}_{i=1}^{n}$ associated with the variational factors that depend on $\mathbf{v}_k$.

Based on the above model, we compare the performance of LeGrad and LdGrad when simultaneously optimizing $V$ and $W$ for a small set of 100 random binarized MNIST digits [17]. The evolution of the instantaneous bound for $H = 40$ hidden units can be seen in Figure 2(a), where once again LeGrad shows superior performance and increased stability.

In the second series of experiments we consider a more complex sigmoid belief network where the prior $p(\mathbf{x})$ over the hidden units becomes a fully connected distribution parametrized by an

Table 1: NLL scores in the test data for the binarized MNIST dataset. The left part of the table shows results based on sigmoid belief nets (SBN) constructed and trained based on the approach from [10], denoted as NVIL, or by using the LeGrad algorithm. The right part of the table gives the performance of alternative state of the art models (reported in Table 1 in [10]).

| SBN | Dim | Test NLL | Model | Dim | Test NLL |
|---|---|---|---|---|---|
| NVIL | 200-200 | 99.8 | FDARN | 400 | 96.3 |
| NVIL | 200-200-200 | 96.7 | NADE | 500 | 88.9 |
| NVIL | 200-200-500 | 97.0 | DARN | 400 | 93.0 |
| LeGrad | 200 | 96.0 | RBM(CD3) | 500 | 105.5 |
| LeGrad | 300 | 95.1 | RBM(CD25) | 500 | 86.3 |
| LeGrad | 500 | 94.9 | MOB | 500 | 137.6 |

additional set of $K(K+1)/2$ model weights (see Supplementary Material). Such a model can better capture the dependence structure of the hidden units and provide a good density estimator for high dimensional data. We trained this model using the $5 \times 10^4$ training examples of the binarized MNIST by using mini-batches of size 100 and assuming different numbers of hidden units: $H = 200, 300, 500$. Table 1 provides negative log likelihood (NLL) scores for LegGrad and several other methods reported in [10]. Notice that for LeGrad the NLLs are essentially variational upper bounds of the exact NLLs obtained by Monte Carlo approximation of the variational bound (an estimate also considered in [10]). From Table 1 we can observe that LeGrad outperforms the advanced NVIL technique proposed in [10]. Finally, Figure 2(b) and 2(c) displays model weights and few examples of digits generated after having trained the model with $H = 200$ units, respectively.

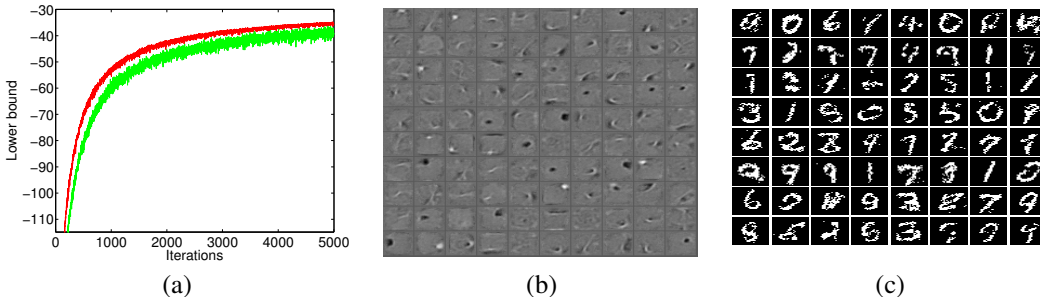

(a)  (b)  (c)

Figure 2: (a) LeGrad (red) and LdGrad (green) convergence for the SBN model on a single mini-batch of 100 MNIST digits. (b) Weights $W$ (filters) learned by LeGrad when training an SBN with $H = 200$ units in the full MNIST training set. (c) New digits generated from the trained model.

## 5 Discussion

Local expectation gradients is a generic black box stochastic optimization algorithm that can be used to maximize objective functions of the form $\mathbf{E}_{q_{\mathbf{v}}(\mathbf{x})}[f(\mathbf{x})]$, a problem that arises in variational inference. The idea behind this algorithm is to exploit the conditional independence structure of the variational distribution $q_{\mathbf{v}}(\mathbf{x})$. Also this algorithm is mostly related to stochastic optimization schemes that make use of the log derivative trick that has been invented in reinforcement learning [3, 21, 13] and has been recently used for variational inference [12, 14, 10]. The approaches in [12, 14, 10] can be thought of as following a global sampling strategy, where multiple samples are drawn from $q_{\mathbf{v}}(\mathbf{x})$ and then variance reduction is built a posteriori in a subsequent stage through the use of control variates. In contrast, local expectation gradients reduce variance by directly changing the sampling strategy, so that instead of working with a global set of samples drawn from $q_{\mathbf{v}}(\mathbf{x})$, the strategy now is to exactly marginalize out the random variable that has the largest influence on a specific gradient of interest while using a single sample for the remaining random variables.

We believe that local expectation gradients can be applied to a great range of stochastic optimization problems that arise in variational inference and in other domains. Here, we have demonstrated its use for variational inference in logistic regression and sigmoid belief networks.

## Footnotes

[1]Notice that $q(x_i|\text{mb}_i^{(t)}) \propto h(x_i, \text{mb}_i^{(t)})q_{v_i}(x_i|\text{pa}_i^{(t)})$ for some non-negative function $h(\cdot)$.

[2]The exact value of the two integrals is the same. However, approximation of these two integrals based on numerical integration will typically not give the same value.

[3]As discussed in [19], there are multiple unbiased sample-based estimators of (4), and using (5) directly tends to have a large variance. We use instead the estimator given by eq. (8) in [19]. Though other estimators with even lower variance exist, we restrict ourselves to those with the same scalability as the proposed LeGrad, requiring at most $\mathcal{O}(S|\mathbf{v}|)$ computation per gradient estimation.

[4]Gaussian quadrature with $K$ grid points integrates exactly polynomials up to $2K - 1$ degree.

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
