[Supplementary Material]

# Supplementary Material: Local Expectation Gradients for Black Box Variational Inference

In this Supplementary Material we include the proofs related to the statements in Section 3.1 of the main paper (Appendix A). We also provide further details, such as updates equations and additional plots, for the application in sigmoid belief networks (Appendix B).

## A   Proofs

**Proposition 1.** The variance of the stochastic gradient $\sum_{x_i} \nabla_{v_i} q_{v_i}(x) f(\mathbf{x}_{\backslash i}, x_i)$ can be written as

$$\sum_{x_i, x_i'} \nabla_{v_i} q_{v_i}(x_i) \nabla_{v_i} q_{v_i}(x_i') \text{Cov}(x_i, x_i'). \tag{1}$$

*Proof:* By starting from the basic definition of the variance of the stochastic estimate we have

$$\mathbb{E}_{q(\mathbf{x}_{\backslash i})} \left[ \left( \sum_{x_i} \nabla_{v_i} q_{v_i}(x_i) f(\mathbf{x}_{\backslash i}, x_i) - \sum_{x_i} \nabla_{v_i} q_{v_i}(x_i) m(x_i) \right)^2 \right]$$

$$= \mathbb{E}_{q(\mathbf{x}_{\backslash i})} \left[ \left( \sum_{x_i} \nabla_{v_i} q_{v_i}(x_i) \left( f(\mathbf{x}_{\backslash i}, x_i) - m(x_i) \right) \right)^2 \right]$$

$$= \mathbb{E}_{q(\mathbf{x}_{\backslash i})} \left[ \sum_{x_i, x_i'} \nabla_{v_i} q_{v_i}(x_i) \nabla_{v_i} q_{v_i}(x_i') \left( f(\mathbf{x}_{\backslash i}, x_i) - m(x_i) \right) \left( f(\mathbf{x}_{\backslash i}, x_i') - m(x_i') \right) \right]$$

$$= \sum_{x_i, x_i'} \nabla_{v_i} q_{v_i}(x_i) \nabla_{v_i} q_{v_i}(x_i') \mathbb{E}_{q(\mathbf{x}_{\backslash i})} \left[ \left( f(\mathbf{x}_{\backslash i}, x_i) - m(x_i) \right) \left( f(\mathbf{x}_{\backslash i}, x_i') - m(x_i') \right) \right]$$

$$= \sum_{x_i, x_i'} \nabla_{v_i} q_{v_i}(x_i) \nabla_{v_i} q_{v_i}(x_i') \text{Cov}(x_i, x_i'). \tag{2}$$

**Proposition 2.** If $\text{Cov}(x_i, x_i') = c$ for all $x_i$ and $x_i'$ then the variance in (1) is equal to zero.

*Proof:* We have

$$\sum_{x_i, x_i'} \nabla_{v_i} q_{v_i}(x_i) \nabla_{v_i} q_{v_i}(x_i') \text{Cov}(x_i, x_i')$$

$$= c \sum_{x_i, x_i'} \nabla_{v_i} q_{v_i}(x_i) \nabla_{v_i} q_{v_i}(x_i')$$

$$= c \left( \sum_{x_i} \nabla_{v_i} q_{v_i}(x_i) \right)^2$$

$$= 0, \tag{3}$$

where we used the fact that $\sum_{x_i} \nabla_{v_i} q_{v_i}(x_i) = 0$.

**Proposition 3.** Given that we can reparametrize $x_i$ as described in the main paper (and all differentiability conditions hold), the gradient $\int \nabla_{v_i} q_{v_i}(x) f(\mathbf{x}_{\backslash i}, x_i) dx_i$ can be equivalently written as

$$\int q_i(z_i) \nabla_{v_i} f(\mathbf{x}_{\backslash i}, g(v_i, z_i)) dz_i, \quad \mathbf{x}_{\backslash i} \sim q(\mathbf{x}_{\backslash i}). \tag{4}$$

*Proof:* We have that

$$\int \nabla_{v_i} q_{v_i}(x) f(\mathbf{x}_{\backslash i}, x_i) dx_i$$
$$= \nabla_{v_i} \left( \int q_{v_i}(x) f(\mathbf{x}_{\backslash i}, x_i) dx_i \right)$$
$$= \nabla_{v_i} \left( \int q_i(z_i) f(\mathbf{x}_{\backslash i}, g(v_i, z_i)) dz_i \right)$$
$$= \int q_i(z_i) \nabla_{v_i} f(\mathbf{x}_{\backslash i}, g(v_i, z_i)) dz_i. \tag{5}$$

**Proposition 4.** If the function $f(\mathbf{x})$ factorizes as $f(\mathbf{x}_{\backslash i}, x_i) = f_i(x_i) + f_{\backslash i}(\mathbf{x}_{\backslash i})$ then $\text{Cov}(x_i, x_i') = c$ for some constant $c$.

*Proof:* First of all observe that
$$m(x_i) = \mathbb{E}_{q(\mathbf{x}_{\backslash i})}[f(\mathbf{x}_{\backslash i}, x_i)] = \mathbb{E}_{q(\mathbf{x}_{\backslash i})}[f_i(x_i) + f_{\backslash i}(\mathbf{x}_{\backslash i})]$$
$$= f_i(x_i) + \mathbb{E}_{q(\mathbf{x}_{\backslash i})}[f_{\backslash i}(\mathbf{x}_{\backslash i})] = f_i(x_i) + k. \tag{6}$$
Now by starting from the definition of $\text{Cov}(x_i, x_i')$ we have
$$\text{Cov}(x_i, x_i') = \mathbb{E}_{q(\mathbf{x}_{\backslash i})}[(f(\mathbf{x}_{\backslash i}, x_i) - m(x_i))(f(\mathbf{x}_{\backslash i}, x_i') - m(x_i'))]$$
$$= \mathbb{E}_{q(\mathbf{x}_{\backslash i})}[(f_i(x_i) + f_{\backslash i}(\mathbf{x}_{\backslash i}) - f_i(x_i) - k)(f_i(x_i') + f_{\backslash i}(\mathbf{x}_{\backslash i}) - f_i(x_i') - k)]$$
$$= \mathbb{E}_{q(\mathbf{x}_{\backslash i})}[(f_{\backslash i}(\mathbf{x}_{\backslash i}) - k)(f_{\backslash i}(\mathbf{x}_{\backslash i}) - k)]$$
$$= c. \tag{7}$$

# B  Further details about the application to sigmoid belief networks with fully connected hidden units

Consider a sigmoid belief network having a fully connected hidden structure
$$p(\mathbf{y}, \mathbf{x}) = \left( \prod_{d=1}^{D} p(y_d|\mathbf{x}) \right) p(x_1) \prod_{k=2}^{K} p(x_k|x_1, \ldots, x_{k-1}),$$
where
$$p(\mathbf{y}|\mathbf{x}, W) = \prod_{d=1}^{D} \left[ \sigma(\mathbf{w}_d^\top \mathbf{x}) \right]^{y_d} \left[ 1 - \sigma(\mathbf{w}_d^\top \mathbf{x}) \right]^{1-y_d} \tag{8}$$
and
$$p(x_1 = 1) = \sigma(a_{10})$$
$$p(x_k = 1|x_1, \ldots, x_{k-1}) = \sigma \left( a_{k0} + \sum_{j=1}^{k-1} a_{kj} x_j \right).$$

The lower bound is written as sum of terms so that each term is
$$\mathcal{F}_i = \sum_{\mathbf{x}_i} q_{\mathbf{v}_i}(\mathbf{x}_i) \left[ \sum_{d=1}^{D} \left[ y_{id} \log \sigma(\mathbf{w}_d^\top \mathbf{x}_i) + (1 - y_{id}) \log(1 - \sigma(\mathbf{w}_d^\top \mathbf{x}_i)) \right] \right.$$
$$+ \sum_{k=1}^{K} \left[ x_{ik} \log \sigma(a_{k0} + \sum_{j=1}^{k-1} a_{kj} x_{ij}) + (1 - x_{ik}) \log(1 - \sigma(a_{k0} + \sum_{j=1}^{k-1} a_{kj} x_{ij})) \right] \right]$$
$$- \sum_{k=1}^{K} \left[ \sigma(v_{ik}) \log \sigma(v_{ik}) + (1 - \sigma(v_{ik})) \log(1 - \sigma(v_{ik})) \right]$$
$$= \sum_{\mathbf{x}_i} q_{\mathbf{v}_i}(\mathbf{x}_i) \left[ - \sum_{d=1}^{D} \log(1 + e^{-\tilde{y}_{id} \mathbf{w}_d^\top \mathbf{x}_i}) - \sum_{k=1}^{K} \log \left( 1 + e^{-\tilde{x}_{ik}(a_{k0} + \sum_{j=1}^{k-1} a_{kj} x_{ij})} \right) \right]$$
$$- \sum_{k=1}^{K} \left[ \sigma(v_{ik}) \log \sigma(v_{ik}) + (1 - \sigma(v_{ik})) \log(1 - \sigma(v_{ik})) \right]. \tag{9}$$

The local expectation gradients assuming a fully factorized variational distribution $q_{\mathbf{v}}(\mathbf{x}) = \prod_{i,k} q_{v_{ik}}(x_{ik})$ (the gradients in the case of the variational recognition model, that was used in the experiments, have almost identical form) to approximate the gradient over $v_{ik}$ is derived as

$$
\begin{aligned}
\nabla_{v_{ik}}\mathcal{F}_i \approx & \sum_{x_{ik}} q_{v_{ik}}(x_{ik})\nabla_{v_{ik}}\log q_{v_{ik}}(x_{ik}) \left[ -\sum_{d=1}^{D}\log(1+e^{-\widetilde{y}_{id}\mathbf{w}_d^\top\{\mathbf{x}_{i\backslash k}^{(t)},x_{ik}\}}) \right.\\
& -\sum_{\ell=1}^{k-1}\log\left(1+e^{-\widetilde{x}_{i\ell}^{(t)}\left(a_{k0}+\sum_{j=1}^{\ell-1}a_{\ell j}x_{ij}^{(t)}\right)}\right) - \log\left(1+e^{-\widetilde{x}_{ik}\left(a_{k0}+\sum_{j=1}^{k-1}a_{kj}x_{ij}^{(t)}\right)}\right)\\
& \left. -\sum_{\ell=k+1}^{K}\log\left(1+e^{-\widetilde{x}_{i\ell}^{(t)}\left(a_{k0}+\sum_{j=1}^{k-1}a_{\ell j}x_{ij}^{(t)}+a_{\ell k}x_{ik}+\sum_{j=k+1}^{\ell-1}a_{\ell j}x_{ij}^{(t)}\right)}\right)\right]\\
& +\sigma_{ik}(1-\sigma_{ik})\log\frac{1-\sigma_{ik}}{\sigma_{ik}}\\
= & \;\sigma_{ik}(1-\sigma_{ik})\left[ -\sum_{d=1}^{D}\log(1+e^{-\widetilde{y}_{id}\mathbf{w}_d^\top\{\mathbf{x}_{i\backslash k}^{(t)},x_{ik}=1\}}) \right.\\
& -\sum_{\ell=1}^{k-1}\log\left(1+e^{-\widetilde{x}_{i\ell}^{(t)}\left(a_{k0}+\sum_{j=1}^{\ell-1}a_{\ell j}x_{ij}^{(t)}\right)}\right) - \log\left(1+e^{-a_{k0}-\sum_{j=1}^{k-1}a_{kj}x_{ij}^{(t)}}\right)\\
& \left. -\sum_{\ell=k+1}^{K}\log\left(1+e^{-\widetilde{x}_{i\ell}^{(t)}\left(a_{k0}+\sum_{j=1}^{k-1}a_{\ell j}x_{ij}^{(t)}+a_{\ell k}+\sum_{j=k+1}^{\ell-1}a_{\ell j}x_{ij}^{(t)}\right)}\right)\right]\\
& +\sigma_{ik}(1-\sigma_{ik})\left[ \sum_{d=1}^{D}\log(1+e^{-\widetilde{y}_{id}\mathbf{w}_d^\top\{\mathbf{x}_{i\backslash k}^{(t)},x_{ik}=0\}}) \right.\\
& \sum_{\ell=1}^{k-1}\log\left(1+e^{-\widetilde{x}_{i\ell}^{(t)}\left(a_{k0}+\sum_{j=1}^{\ell-1}a_{\ell j}x_{ij}^{(t)}\right)}\right) + \log\left(1+e^{a_{k0}+\sum_{j=1}^{k-1}a_{kj}x_{ij}^{(t)}}\right)\\
& \left. +\sum_{\ell=k+1}^{K}\log\left(1+e^{-\widetilde{x}_{i\ell}^{(t)}\left(a_{k0}+\sum_{j=1}^{k-1}a_{\ell j}x_{ij}^{(t)}+\sum_{j=k+1}^{\ell-1}a_{\ell j}x_{ij}^{(t)}\right)}\right)\right]\\
& +\sigma_{ik}(1-\sigma_{ik})\log\frac{1-\sigma_{ik}}{\sigma_{ik}}, \qquad (10)
\end{aligned}
$$

or

$$
\begin{aligned}
\nabla_{v_{ik}}\mathcal{F}_i \approx \sigma_{ik}(1-\sigma_{ik})&\left[ \sum_{d=1}^{D}\log\frac{1+e^{-\widetilde{y}_{id}\mathbf{w}_d^\top\{\mathbf{x}_{i\backslash k}^{(t)},x_{ik}=0\}}}{1+e^{-\widetilde{y}_{id}\mathbf{w}_d^\top\{\mathbf{x}_{i\backslash k}^{(t)},x_{ik}=1\}}} + \log\frac{1+e^{a_{k0}+\sum_{j=1}^{k-1}a_{kj}x_{ij}^{(t)}}}{1+e^{-a_{k0}-\sum_{j=1}^{k-1}a_{kj}x_{ij}^{(t)}}} \right.\\
& \left. +\sum_{\ell=k+1}^{K}\log\frac{1+e^{-\widetilde{x}_{i\ell}^{(t)}\left(a_{\ell0}+\sum_{j=1}^{k-1}a_{\ell j}x_{ij}^{(t)}+\sum_{j=k+1}^{\ell-1}a_{\ell j}x_{ij}^{(t)}\right)}}{1+e^{-\widetilde{x}_{i\ell}^{(t)}\left(a_{\ell0}+\sum_{j=1}^{k-1}a_{\ell j}x_{ij}^{(t)}+a_{\ell k}+\sum_{j=k+1}^{\ell-1}a_{\ell j}x_{ij}^{(t)}\right)}} + \log\frac{1-\sigma_{ik}}{\sigma_{ik}}\right]. \quad (11)
\end{aligned}
$$

The instantaneous gradient wrt $W$ is:

$$
\nabla_{\mathbf{w}_k}\mathcal{F}_i \approx \sum_{x_{ik}} x_{ik}\mathbf{y}_i - x_{ik}\sigma\left(\mathbf{w}_0 + \sum_k x_{ik}\mathbf{w}_k\right). \qquad (12)
$$

And the instantaneous gradient wrt $A$ is:

$$
\nabla_{a_{kj}}\mathcal{F}_i \approx \sum_{x_{ik},x_{ij}} x_{ik}x_{ij} - x_{ik}\sigma\left(a_{k0} + \sum_k x_{ik}a_{kj}\right). \qquad (13)
$$

Finally, the following figures display model weights and few examples of the SBN output probabilities (based on which we can generated binary digits) after having trained the model with $H = 300, 500$ units.

Figure 1: The left panel shows some of the parameters weights $W$ (filters) learned by LeGrad when training a sigmoid belief net with $H = 300$ units. The right panel shows several examples of final probability values of the SBN when we simulate the hidden states in the trained model.

Figure 2: The left panel shows some of the parameters weights $W$ (filters) learned by LeGrad when training a sigmoid belief net with $H = 500$ units. The right panel shows several examples of final probability values of the SBN when we simulate the hidden states in the trained model.