[Reviews · NeurIPS 2015]

Submitted by Assigned_Reviewer_1

SUMMARY This paper presents an idea that would be very interesting to NIPS readers. It presents a preliminary analysis of an estimator of a quantity that every variational inference researcher would want to know about. The mathematical exposition is good: the authors promise more technical and theoretical studies down the road. The experimental study is the weakest point of the paper, where they miss the opportunity to explore practical aspects of their estimator (specifically, other integration techniques beyond Gaussian quadrature). Overall it is a good paper.

QUALITY The paper presents a new method in a careful and well-paced manner. The mathematical development is good. However, the experimental section could use a bit more work. My main comment is that I would have rather had the authors study alternative ways of computing (or approximating) the internal expectation in Equation (9). Instead, they study two uninteresting toy models in Sections 4.1 and 4.2, where it is not clear what insight they expect the reader to extract. The experimental study in Section 4.3 is good, because it addresses a case where ReGrad is not applicable. This is important because there appears to be little reason to use LeGrad for differentiable probability models.

CLARITY The paper is well written and accessible.

I am not particularly keen on how Section 3.1 is structured. It's not clear to me why the authors have bundled a time-complexity study with the (in my opinion) much more interesting connection to Gibbs sampling. I would rather prefer that they minimize the time-complexity paragraph and push it to the experimental section; then, they could devote this space in the narrative to the connection to Gibbs (and maybe even expand on it some more).

Figure 2: caption describes (a) (a) (c). should be (a) (b) (c).

Figure 3 also has some caption issues.

ORIGINALITY The authors propose a new idea for estimating the gradient of an expectation that pops up in many of the state-of-the-art variational inference techniques proposed in the last few years. This paper should have no trouble on the originality front.

SPECIFICS line 151: extra comma in equation, needs reformatting.

line 196: ... that IT has with ...

line 250: ... we consider --a-- sigmoid belief networks ...
Summary: This paper presents a preliminary analysis of a gradient estimator for variational inference. The topic is interesting, the solution is original, the maths are good, and the empirical study is okay. Presentation is good, small errors can be corrected for the camera-ready. [Update post-discussion] My concerns about the empirical study have increased, especially around the practicality of this method and its connection to control variate design compared to Paisley 2012 and Ranganath 2014.

Submitted by Assigned_Reviewer_2

The authors discuss a stochastic gradient method when the variational objective function is not tractable. They key proposal seems to be Eq 9.

This is said to be improving the previous methods somehow, but it isn't clear how this is any different from, e.g., Paisley, et al. (2012). This seems to be simply writing out the factorizations more explicitly that the previous works didn't bother to do. Is there something more to it than this?

Other comments:

Fig 1b: It looks like the proposed method performs the worst on this experiment.

Sec 4.2: The authors should have compared using a control variate in addition to LdGrad, which is the method of Paisley, et al. without a control variate. It seems like a very relevant aspect of the Paisley, et al. paper is being deliberately ignored in these experiments. For example, the variance of LdGrad is much larger than LeGrad, but that was acknowledged as typically the case in previous work, which is why straightforward control variates have been used.

Fig 2c: The proposed algorithm is converging much more slowly. This seems to defeat the purpose of the method.

Comment: The paper refers to this in several places as a "stochastic variational inference" algorithm. While it is stochastic and variational inference, that expression has been established to mean something very different from what is being done here. While this is a stochastic optimization approach to variational inference, it should be shortened to something else.
Summary: The authors propose a stochastic gradients method for intractable VI objectives. The main contribution of the paper is unclear to me and the comparisons are somewhat weak and not entirely representative of what has been proposed in this area.

Author Feedback
Author rebuttal: We are grateful to all reviewers for their comments and we will address them in the updated version of the paper. Next, we wish to give some important clarifications regarding the novelty of our method (as compared to Paisley et al. (2012)) and the specific LdGrad method that we used in the comparisons.

This proposal is different from Paisley et al. (2012) in that
a) it performs exact marginalization over some random variables when computing the unbiased approximation of the gradient
b) the random variables over which exact marginalization is performed are different for each component of the gradient
c) for each gradient component, the random variable that has the largest influence on it, is the one exactly marginalized out
This is a concrete marginalization strategy that adds little overhead and reduces variance by sampling where it matters most. The samples in Paisley et al. (2012) are uniform/global samples that do not follow the above strategy.

A very important clarification: LdGrad is *not* the method of Paisley, et al. without a control variate. The LdGrad method we implemented is based on the eq. (8) in the paper "Tim Salimans and David A. Knowles. On Using Control Variates with Stochastic Approximation for Variational Bayes and its Connection to Stochastic Linear Regression, January 2014." We mentioned this in our paper in lines 246 (footnote 1) and line 350. This method uses variance reduction from a regression perspective and works similarly to a typical use of control variates (the derivative of log q(x) signal is centered which results in significant variance reduction in the stochastic gradient). Without this variance reduction process the LdGrad method (i.e. implemented naively based on the estimate from eq (5) in our submitted paper) does not work at all as the gradients have extremely high variance. We want to further clarify that we have implemented this particular version of LdGrad from Salimans and Knowles because it is both black-box and scalable (it scales linearly with the dimensionality of the latent variables/parameters), so that it is comparable to the generality and scope of our method which is also black-box and has linear complexity. Of course other versions of LdGrad, such the one originally proposed by Paisley et. al., are possible but they are not so "black-box" because use control variate functions that depend on the probabilistic model at hand, i.e. the user (who might not be an expert) needs to design these functions in a per-model basis which can be a very difficult task.

So we hope that the above clarifies the fact that **we do use variance reduction within LdGrad (otherwise this method cannot work in practice)** and we will update the paper accordingly to make this point clear.